# A Dynamic Prompt-tuning Method for Data Augmentation with Associated Knowledge

**Qianqian Qi , Qiming Bao,**[*] **Alex Yuxuan Peng, Jiamou Liu, Michael Witbrock**
Strong AI Lab
School of Computer Science
University of Auckland
{qqi518, qiming.bao, ypen260}@aucklanduni.ac.nz
{jiamou.liu, m.witbrock }@auckland.ac.nz

## Abstract

Transformer-based pretrained language models (PLMs) have shown to pre-learn rich prior knowledge. To assist data-to-text task, we propose a new dynamic prompt tuning method, DPTAK, to retrieve knowledge from a PLM that is associated with individual data-text pairs. Our method increases the diversity of the training examples without the need to manually collecting and labelling data. When applied on GPT-2, DPTAK outperforms baseline models in several well-studied data-to-text and text-to-data datasets such as E2E, WebNLG, DART.

## 1 Introduction and Related work

*Data-to-text* task is important in natural language processing (NLP) with wide-ranging applications such as biography generation (Lebret et al., 2016) and question answering (Shi & Lin, 2019). In these tasks, Transformer-based (Vaswani et al., 2017) pretrained language models (PLMs) have achieved state-of-the-art performance. Nevertheless, obtaining a sufficiently large dataset that is necessary to tune a PLM to high accuracy in a domain-specific setting remains a bottleneck. *Data augmentation* (DA) addresses this bottleneck by enriching existing datasets with additional synthetic data, through which the model enhances its performance. Leveraging the superior language generation abilities of PLMs, an important class of DA methods employ PLM to produce synthetic data. Compared with traditional methods, these tools may produce syntatically rich, yet coherent and consistent synthetic dataset (Kumar et al., 2020). Given an input corpus, if one is able to elicit knowledge that are not present in but implicitly-linked to the input corpus, then such knowledge can be used to construct an augmented dataset. With this inspiration, we propose a new DA paradigm, namely Dynamic Prompt Tuning Method with Associated Knowledge (DPTAK), for the data-to-text task. The main challenge lies in "controlled elicitation", i.e., allowing a PLM to output knowledge that is not present in but implicitly-linked with the input corpus. We adopt *prompt tuning* (Liu et al., 2021) as the main methodology which reformulates downstream tasks into the form of the pretrained task while tuning the PLM, thereby helping the PLM to recall its pre-learned knowledge.

## 2 Method

The proposed Dynamic Prompt Tuning Method with Associated Knowledge (DPTAK) contains three phases: 1. Associated knowledge retrieval: This phase extracts associated knowledge from PLM using the prompt tuning method. The associate knowledge is generated in the form of sentences. 2. Data distillation: The generated associated knowledge is usually noisy and contains redundancy. This phase thus aims to trim the generated knowledge through filtering out poor-quality data. 3. Pairwise data acquisition: This phase produces the required data-text samples for the downstream tasks using the generated knowledge.

We select GPT-2 as the underlying model to retrieve the associated knowledge given input-label pairs. Our prompt contains two parts: a natural language enquiry and a semantic representation

---

[*]Corresponding author

Table 1: Experimental results on E2E, WebNLG, and DART for the data-to-text task.

| Dataset | Model | BLEU | Rouge | Perplexity (PPL)↓ | BERTScore | Readability | Self-BLEU↓ | Coverage |
|---|---|---|---|---|---|---|---|---|
| E2E | T5 | 67.22 | 60.59 | 5.65 | 95.39 | 25.16 | **88.31** | **88.26** |
| | T5-DPTAK | **67.94** | **60.64** | **1.99** | **95.46** | 24.53 | 88.35 | 87.70 |
| | GPT-2 | 65.71 | 60.25 | 2.59 | **95.75** | 25.87 | 86.45 | 86.25 |
| | GPT-2-NLPAUG | 65.21 | 59.80 | 4.60 | 95.25 | 27.45 | 85.56 | 86.55 |
| | GPT-2-SSMBA | 63.40 | 60.04 | 2.89 | 95.25 | 30.21 | **85.05** | 86.74 |
| | GPT-2-DPTAK | 66.08 | 60.11 | 2.54 | 95.34 | 27.79 | 86.42 | **87.13** |
| WebNLG | T5 | 56.21 | 63.65 | 1.67 | **95.30** | 40.54 | 87.11 | 79.00 |
| | T5-DPTAK | **56.61** | **63.91** | **1.64** | 95.29 | 41.15 | 87.21 | **80.00** |
| | GPT-2 | 41.11 | 53.23 | 1.89 | **92.95** | 43.27 | 80.03 | 56.29 |
| | GPT-2-NLPAUG | 42.47 | **53.51** | 1.78 | 92.24 | 44.71 | 80.04 | 56.23 |
| | GPT-2-SSMBA | 42.40 | 53.17 | 1.80 | 92.25 | 45.52 | **79.99** | 55.70 |
| | GPT-2-DPTAK | **42.91** | 53.47 | 1.71 | 92.29 | 43.57 | 80.25 | 56.20 |
| DART | T5 | 48.17 | 61.58 | 2.80 | **95.05** | 39.18 | 88.05 | 86.96 |
| | T5-DPTAK | **48.55** | **61.72** | **1.81** | 95.03 | 40.54 | **87.97** | **87.11** |
| | GPT-2 | 43.22 | 57.95 | 2.56 | **94.61** | 40.18 | 84.11 | 81.74 |
| | GPT-2-NLPAUG | 42.26 | 57.05 | 2.86 | 93.86 | 36.34 | **83.14** | 80.16 |
| | GPT-2-SSMBA | 42.26 | 57.01 | 2.95 | 93.94 | 39.39 | 83.32 | 79.92 |
| | GPT-2-DPTAK | **44.11** | **58.45** | 2.05 | 94.09 | 41.52 | 84.76 | 81.79 |

enquiry. The natural language enquiry is the linearized structured data. The semantic representation enquiry is generated from the label of the sample, and it is used to help GPT-2 retrieve associated knowledge from continual hidden space. We adopt Paraphrase-DistilRoBERTa (Sanh et al., 2019) which provides superior semantic representation $pr$ of the labels. A prompt linear layer takes the semantic representation $pr$ as input and outputs a hidden representation vector $h_{\mathrm{mem}}$. This process of DPTAK is more formally described in Algorithm 1, and more details are described in the Appendix.

---

**Algorithm 1** DPTAK

**Input:** Training Dataset $D_{\mathcal{D}} = \{x_i, y_i\}_k^1$,
Generative model $PLM_{\mathrm{gen}}$,
Semantic embedding model $PLM_{\mathrm{sem}}$
Pairwise data generation model $PLM_{\mathrm{pair}}$
**Output:** Augmented Dataset $D_{\mathrm{aug}} = \{\hat{x}_i, \hat{y}_i\}_s^1$
1 **for** $\{x_i, y_i\} \in D_{\mathcal{D}}$ **do**
2     Obtain the semantic representation $pr = PLM_{\mathrm{sem}}(y_i)$
     Prepare $PLM_{\mathrm{gen}}$ model input $c_i = concat(x_i, \langle BOS \rangle, y_i)$
     Memory vector $h_{\mathrm{mem}} = FullyConnected(pr)$
     Pre-computed hidden-states $(key, value) = (h_{\mathrm{mem}}, h_{\mathrm{mem}})$
     $\tilde{K}, \tilde{V} = concat(h_{\mathrm{mem}}, K), concat(h_{\mathrm{mem}}, V)$
     Computing attentions $Attention(Q, \tilde{K}, \tilde{V}) = Softmax\left(\frac{Q\tilde{K}^{\mathsf{T}}}{\sqrt{d}}\right)\tilde{V}$
     $Pred_i = PLM_{\mathrm{gen}}(c_i, Attention(Q, \tilde{K}, \tilde{V}))$
     $masked\_pred_i = mask(pred_i)$
     Fine-tune $PLM_{\mathrm{gen}}$ by minimizing $CrossEntropyLoss(c_i, masked_p red_i)$
3 Obtain fine-tuned model $PLM_{retrieval}$
   Initialise $D_{\mathrm{gen}} = \emptyset$
   **for** $\{x_i, y_i\} \in D_{\mathcal{D}}$ **do**
4     $D_{\mathrm{gen}} = D_{\mathrm{gen}} \cup \{PLM_{retrieval}(x_i, y_i)\}$
5 Obtain prediction samples $D_{\mathrm{gen}}$
   Get distilled dataset $D_{distil} = DataDistillation(D_{\mathrm{gen}})$
   Get data-text pairs $D_{\mathrm{aug}} = PLM_{\mathrm{pair}}(D_{distil})$
   return $D_{\mathrm{aug}}$

---

## 3 EXPERIMENTS AND CONCLUSION

We evaluated DPTAK on three different datasets: E2E (Novikova et al., 2017), WebNLG (Gardent et al., 2017) and DART (Nan et al., 2021). We compared our method to five different baselines: standard fine-tuning of PLMs (T5 and GPT-2) without any data augmentation, NLPAUG and SSMBA (Ng et al.). The evaluation metrics include: BLEU, ROUGE, semantics (BERTScore), diversity (Self-BLEU), Coverage (Jolly et al., 2021), Perplexity (PPL) (Chen & Goodman, 1999) and readability (Coleman–Liau). More details of the evaluation methods and baselines are described in the Appendix. Table 1 shows the results of data-to-text task on E2E, WebNLG and DART datasets.

T5-DPTAK achieved better performance than T5 on most of the evaluation metrics. GPT-2-DPTAK achieved higher BLEU scores than other baseline models when GPT-2 was used instead of T5.

In summary, our contributions are three-fold: (1) We propose a new DA paradigm that aims to elicit pre-learned knowledge from PLMs for data-to-text task. (2) We introduce a new dynamic prompt tuning method to achieve controlled elicitation from PLMs. The method utilises both of the natural text and learned semantic representations. (3) We verify that the proposed DA approach enhances the performance of data-to-text task.

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

## A APPENDIX

### A.1 EVALUATION METRICS

Other than two commonly used automatic evaluation metrics, such as BLEU and ROUGE, we also conducted evaluation on the semantics, diversity and readability of generated texts. BERTScore is a

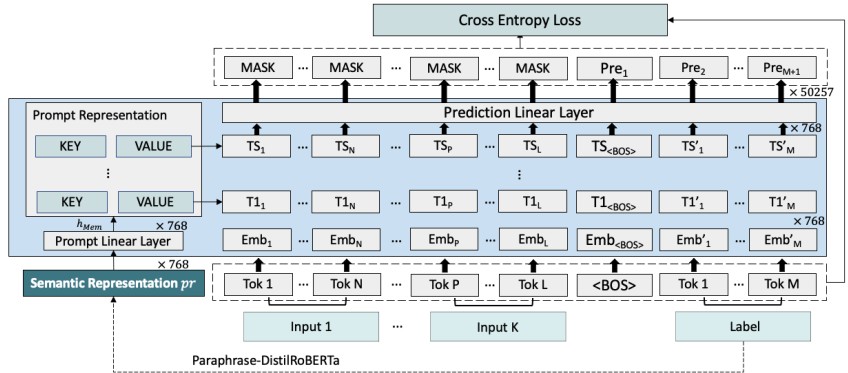

Figure 1: Training process of associate knowledge retrieval framework.

metric for evaluating semantic similarity between two text inputs. Self-BLEU evaluates the diversity of the generated texts. Coleman–Liau index measures the readability of texts. The output indicates the US grade level required to understand the texts. The Coverage index follows the concept of hard coverage from existing work Jolly et al. (2021). Coverage in the data-to-text task indicates how much information in the structured data is covered by the text. It is calculated as a ratio of the number of phrases from the structured data that are included in the generated text, over the total number of phrases in the structured data. Coverage in the text-to-data task measures the amount of information in the text is contained in the structured data. Finally, Perplexity (PPL) Chen & Goodman (1999) is a common metric used to measure how well a language model predicts a sample of texts.

## A.2 BASELINE MODELS

or the data-to-text task, we compared our method to four different baselines: standard fine-tuning of PLMs (T5 and GPT-2) without any data augmentation, NLPAUG (https://pypi.org/project/nlpaug/), and SSMBA Ng et al.. BART-paraphraser is a BART-based model fine-tuned to do paraphrasing. NLPAUG is a python toolbox for textual data augmentation. We randomly chose a method provided in NLPAUG: swap, delete, substitute, or insert, and applied it to the given text. SSMBA is a sampling-based data augmentation method that uses a corruption function to perturb the original data distribution and then uses a reconstruction function to reconstruct the sentence as the augmented text. In data-to-text task, these methods were used to augment original texts in the training data. Then we applied the same filtering and data-text pair generation processes as described in Method Section to the augmented texts.

## A.3 DPTAK

**1. Associated Knowledge Retrieval**

Figure 1 describes the training process of our model. We linearise the input data as text sequence and concatenate it with its text description. There are many tokens within each input and the label, and a special token $\langle BOS \rangle$ is used as the delimiter between input and label. We use the dynamic prompt tuning method to get attention scores in Transformer layers. There is a prediction linear layer after Transformer blocks for next token prediction. We mask the predictions that are generated from the input tokens before the $\langle BOS \rangle$ token. Then Cross Entropy is calculated between the model input and the masked output as the Language modeling loss.

**2. Data Distillation** We filter the generated texts using text similarity score (BLEU) and input coverage rate Jolly et al. (2021). If BLEU and input coverage rate are too low, the generated text is likely to deviate too much from the distribution of the original training data. In addition, very high BLEU score may indicate the generated text is almost identical to the label in the training data. In this case, the generated data is unlikely to provide any gain in performance by including it in the training set. Hence, we select the generated texts that are within a BLEU score range and have relatively high input coverage rate.

**3. Pairwise Data Acquisition**    After the first two steps, we apply a generator model to generate pairwise data based on the distilled data. Then we get the new data-text pairs which form the augmented dataset.

