# OpenReview forum: "A Dynamic Prompt-tuning Method for Data Augmentation with Associated Knowledge"
_ICLR.cc/2023/TinyPapers — Submitted to Tiny Papers @ ICLR 2023_

### Official Review · Reviewer_AcBz · 2023-03-20

**Confidence:** 4

**Summary Of Contributions:**

A good idea but have some issues

**Rating:**

Great Start (GS): a submission which meets some of the reviewing criteria but has room for improvement

**Strengths And Weaknesses:**

This paper presents a novel dynamic prompt tuning method, DPTAK, for retrieving knowledge from a PLM that is associated with individual data-text pairs. This method enhances the diversity of training examples without requiring manual data collection and labeling. When applied to GPT-2, DPTAK outperforms baseline models on several well-studied data-to-text and text-to-data datasets, such as E2E, WebNLG, and DART.

Strengths:

1.The paper introduces a new dynamic prompt tuning method, DPTAK, for knowledge retrieval from a PLM associated with individual data-text pairs.

2.The experimental results demonstrate the effectiveness of the proposed method.

Weaknesses:

I have some concerns about the knowledge retrieval part, as it relies on another model, Paraphrase-DistilRoBERTa, for providing semantic representation. Therefore, it is questionable whether this constitutes true knowledge retrieval. Furthermore, the paper could benefit from comparisons with other methods, such as random initialization or prefix tuning.

**Suggested Changes:**

I have some concerns about the knowledge retrieval part, as it relies on another model, Paraphrase-DistilRoBERTa, for providing semantic representation. Therefore, it is questionable whether this constitutes true knowledge retrieval. Furthermore, the paper could benefit from comparisons with other methods, such as random initialization or prefix tuning.

---

### Official Review · Reviewer_1vVQ · 2023-03-29

**Confidence:** 3

**Summary Of Contributions:**

This paper presents Dynamic Prompt Tuning Method with Associated Knowledge (DPTAK), a prompt tuning method that retrieve knowledge associated with individual data-text pairs from pretrained language models. Comprehensive experiments highlight that DPTAK improve the model's performance on data-to-text tasks.

**Rating:**

Clear, Correct, and Reproducible (CCR): a submission which meets the reviewing criteria

**Strengths And Weaknesses:**

## Strengths and weaknesses

#### Clarity

1. Overall the paper is clear and easy to read.
2. Details in the method section can be further clarified if there is more space.

#### Correctness

1. The claims and conclusions are justified by the findings.

#### Reproducibility

1. The paper includes findings from an empirical experiment, but the code and data are not open-sourced.

#### Follows basic requirements

1. This paper follows the basic formatting requirements.

**Suggested Changes:**

## Suggested changes

1. The method section is a bit dense. It would be great if the paper can includes some examples to elucidate the algorithm. Are both natural language enquiry and semantic representation vectors?

---

### Author Response · Authors · 2023-05-30
**Opt-in for archival**

Thanks to reviewers and program chair about your contribution to the review process. I am keen to archive my paper.

---

### Meta-Review · Area_Chair_SByt · 2023-04-06

**Recommendation:** Invite to archive
**Confidence:** 3

**Metareview:**

This paper proposes a dynamic prompt tuning method for retrieving knowledge from a PLM that is associated with individual data-text pairs by enhancing the diversity of training examples. When applied to GPT-2, DPTAK outperforms baseline models on several well-studied data-to-text and text-to-data datasets, such as E2E, WebNLG, and DART.
It's not clear whether true knowledge retrieval is happening since it relies on another model for providing semantic representation. Additionally, the paper could benefit from comparisons with other methods, such as random initialization.

**Summary:**

Nice novel approach in augmenting and generating diverse datasets, but could improve validation

**Comments And Feedback To The Authors:**

Nicely written and well-presented. By adding some additional baselines and addressing the reviewers' suggestion, this paper can be a valuable addition to the literature.

**Reason For Not Giving A Higher Recommendation:**

The comparison with baseline approaches or even random would be good to put this method in context.

**Reason For Not Giving A Lower Recommendation:**

The reviewers agree that the novelty of the paper and the idea is valuable.

---

### Decision · Program_Chairs · 2023-04-07

Invite to archive